# The Noisy Power Method:
# A Meta Algorithm with Applications

**Moritz Hardt**[*]
IBM Research Almaden

**Eric Price**[†]
IBM Research Almaden

## Abstract

We provide a new robust convergence analysis of the well-known power method for computing the dominant singular vectors of a matrix that we call the *noisy power method*. Our result characterizes the convergence behavior of the algorithm when a significant amount noise is introduced after each matrix-vector multiplication. The noisy power method can be seen as a meta-algorithm that has recently found a number of important applications in a broad range of machine learning problems including alternating minimization for matrix completion, streaming principal component analysis (PCA), and privacy-preserving spectral analysis. Our general analysis subsumes several existing ad-hoc convergence bounds and resolves a number of open problems in multiple applications:

**Streaming PCA.** A recent work of Mitliagkas et al. (NIPS 2013) gives a space-efficient algorithm for PCA in a streaming model where samples are drawn from a gaussian spiked covariance model. We give a simpler and more general analysis that applies to arbitrary distributions confirming experimental evidence of Mitliagkas et al. Moreover, even in the spiked covariance model our result gives quantitative improvements in a natural parameter regime. It is also notably simpler and follows easily from our general convergence analysis of the noisy power method together with a matrix Chernoff bound.

**Private PCA.** We provide the first nearly-linear time algorithm for the problem of differentially private principal component analysis that achieves nearly tight worst-case error bounds. Complementing our worst-case bounds, we show that the error dependence of our algorithm on the matrix dimension can be replaced by an essentially tight dependence on the *coherence* of the matrix. This result resolves the main problem left open by Hardt and Roth (STOC 2013). The coherence is always bounded by the matrix dimension but often substantially smaller thus leading to strong average-case improvements over the optimal worst-case bound.

## 1 Introduction

Computing the dominant singular vectors of a matrix is one of the most important algorithmic tasks underlying many applications including low-rank approximation, PCA, spectral clustering, dimensionality reduction, matrix completion and topic modeling. The classical problem is well-understood, but many recent applications in machine learning face the fundamental problem of approximately finding singular vectors in the presence of noise. Noise can enter the computation through a variety of sources including sampling error, missing entries, adversarial corruptions and privacy constraints. It is desirable to have one robust method for handling a variety of cases without the need for ad-hoc analyses. In this paper we consider the *noisy power method*, a fast general purpose method for computing the dominant singular vectors of a matrix when the target matrix can only be accessed through inaccurate matrix-vector products.

---

[*]Email: `mhardt@us.ibm.com`
[†]Email: `ecprice@cs.utexas.edu`

Figure 1 describes the method when the target matrix $A$ is a symmetric $d \times d$ matrix—a generalization to asymmetric matrices is straightforward. The algorithm starts from an initial matrix $X_0 \in \mathbb{R}^{d \times p}$ and iteratively attempts to perform the update rule $X_\ell \to AX_\ell$. However, each such matrix product is followed by a possibly adversarially and adaptively chosen perturbation $G_\ell$ leading to the update rule $X_\ell \to AX_\ell + G_\ell$. It will be convenient though not necessary to maintain that $X_\ell$ has orthonormal columns which can be achieved through a QR-factorization after each update.

---

**Input:** Symmetric matrix $A \in \mathbb{R}^{d \times d}$, number of iterations $L$, dimension $p$
    1. Choose $X_0 \in \mathbb{R}^{d \times p}$.
    2. For $\ell = 1$ to $L$:
        (a) $Y_\ell \leftarrow AX_{\ell-1} + G_\ell$ where $G_\ell \in \mathbb{R}^{d \times p}$ is some perturbation
        (b) Let $Y_\ell = X_\ell R_\ell$ be a QR-factorization of $Y_\ell$
**Output:** Matrix $X_L$

---

Figure 1: Noisy Power Method (NPM)

The noisy power method is a meta algorithm that when instantiated with different settings of $G_\ell$ and $X_0$ adapts to a variety of applications. In fact, there have been a number of recent surprising applications of the noisy power method:

1. Jain et al. [JNS13, Har14] observe that the update rule of the well-known alternating least squares heuristic for matrix completion can be considered as an instance of NPM. This lead to the first provable convergence bounds for this important heuristic.

2. Mitgliakas et al. [MCJ13] observe that NPM applies to a streaming model of principal component analysis (PCA) where it leads to a space-efficient and practical algorithm for PCA in settings where the covariance matrix is too large to process directly.

3. Hardt and Roth [HR13] consider the power method in the context of privacy-preserving PCA where noise is added to achieve differential privacy.

In each setting there has so far only been an ad-hoc analysis of the noisy power method. In the first setting, only local convergence is argued, that is, $X_0$ has to be cleverly chosen. In the second setting, the analysis only holds for the spiked covariance model of PCA. In the third application, only the case $p = 1$ was considered.

In this work we give a completely general analysis of the noisy power method that overcomes limitations of previous analyses. Our result characterizes the global convergence properties of the algorithm in terms of the noise $G_\ell$ and the initial subspace $X_0$. We then consider the important case where $X_0$ is a randomly chosen orthonormal basis. This case is rather delicate since the initial correlation between a random matrix $X_0$ and the target subspace is vanishing in the dimension $d$ for small $p$. Another important feature of the analysis is that it shows how $X_\ell$ converges towards the first $k \leqslant p$ singular vectors. Choosing $p$ to be larger than the target dimension leads to a quantitatively stronger result. Theorem 2.3 formally states our convergence bound. Here we highlight one useful corollary to illustrate our more general result.

**Corollary 1.1.** *Let* $k \leqslant p$. *Let* $U \in \mathbb{R}^{d \times k}$ *represent the top* $k$ *singular vectors of* $A$ *and let* $\sigma_1 \geqslant \cdots \geqslant \sigma_n \geqslant 0$ *denote its singular values. Suppose* $X_0$ *is an orthonormal basis of a random* $p$-*dimensional subspace. Further suppose that at every step of* NPM *we have*

$$5\|G_\ell\| \leqslant \varepsilon(\sigma_k - \sigma_{k+1}) \quad and \quad 5\|U^\top G_\ell\| \leqslant (\sigma_k - \sigma_{k+1})\frac{\sqrt{p} - \sqrt{k-1}}{\tau\sqrt{d}}$$

*for some fixed parameter* $\tau$ *and* $\varepsilon < 1/2$. *Then with all but* $\tau^{-\Omega(p+1-k)} + e^{-\Omega(d)}$ *probability, there exists an* $L = O(\frac{\sigma_k}{\sigma_k - \sigma_{k+1}} \log(d\tau/\varepsilon))$ *so that after* $L$ *steps we have that* $\|(I - X_L X_L^\top)U\| \leqslant \varepsilon$.

The corollary shows that the algorithm converges in the strong sense that the entire spectral norm of $U$ up to an $\varepsilon$ error is contained in the space spanned by $X_L$. To achieve this the result places two assumptions on the magnitude of the noise. The total spectral norm of $G_\ell$ must be bounded by $\varepsilon$ times the separation between $\sigma_k$ and $\sigma_{k+1}$. This dependence on the singular value separation arises even in the classical perturbation theory of Davis-Kahan [DK70]. The second condition is specific to the power method and requires that the noise term is proportionally smaller when projected onto the space spanned by the top $k$ singular vectors. This condition ensures that the correlation between $X_\ell$

and $U$ that is initially very small is not destroyed by the noise addition step. If the noise term has some spherical properties (e.g. a Gaussian matrix), we expect the projection onto $U$ to be smaller by a factor of $\sqrt{k/d}$, since the space $U$ is $k$-dimensional. In the case where $p = k + \Omega(k)$ this is precisely what the condition requires. When $p = k$ the requirement is stronger by a factor of $k$. This phenomenon stems from the fact that the smallest singular value of a random $p \times k$ gaussian matrix behaves differently in the square and the rectangular case.

We demonstrate the usefulness of our convergence bound with several novel results in some of the aforementioned applications.

## 1.1 Application to memory-efficient streaming PCA

In the streaming PCA setting we receive a stream of samples $z_1, z_2, \ldots z_n \in \mathbb{R}^d$ drawn i.i.d. from an unknown distribution $\mathcal{D}$ over $\mathbb{R}^d$. Our goal is to compute the dominant $k$ eigenvectors of the covariance matrix $A = \mathbb{E}_{z \sim \mathcal{D}} zz^\top$. The challenge is to do this in space linear in the output size, namely $O(kd)$. Recently, Mitgliakas et al. [MCJ13] gave an algorithm for this problem based on the noisy power method. We analyze the same algorithm, which we restate here and call SPM:

---

**Input:** Stream of samples $z_1, z_2, \ldots, z_n \in \mathbb{R}^d$, iterations $L$, dimension $p$
    1. Let $X_0 \in \mathbb{R}^{d \times p}$ be a random orthonormal basis. Let $T = \lfloor m/L \rfloor$
    2. For $\ell = 1$ to $L$:
        (a) Compute $Y_\ell = A_\ell X_{\ell-1}$ where $A_\ell = \sum_{i=(\ell-1)T+1}^{\ell T} z_i z_i^\top$
        (b) Let $Y_\ell = X_\ell R_\ell$ be a QR-factorization of $Y_\ell$
**Output:** Matrix $X_L$

---

Figure 2: Streaming Power Method (SPM)

The algorithm can be executed in space $O(pd)$ since the update step can compute the $d \times p$ matrix $A_\ell X_{\ell-1}$ incrementally without explicitly computing $A_\ell$. The algorithm maps to our setting by defining $G_\ell = (A_\ell - A)X_{\ell-1}$. With this notation $Y_\ell = AX_{\ell-1} + G_\ell$. We can apply Corollary 1.1 directly once we have suitable bounds on $\|G_\ell\|$ and $\|U^\top G_\ell\|$.

The result of [MCJ13] is specific to the spiked covariance model. The spiked covariance model is defined by an orthonormal basis $U \in \mathbb{R}^{d \times k}$ and a diagonal matrix $\Lambda \in \mathbb{R}^{k \times k}$ with diagonal entries $\lambda_1 \geqslant \lambda_2 \geqslant \cdots \geqslant \lambda_k > 0$. The distribution $\mathcal{D}(U, \Lambda)$ is defined as the normal distribution $N(0, (U\Lambda^2 U^\top + \sigma^2 \mathrm{Id}_{d \times d}))$. Without loss of generality we can scale the examples such that $\lambda_1 = 1$. One corollary of our result shows that the algorithm outputs $X_L$ such that $\left\|(I - X_L X_L^\top)U\right\| \leqslant \varepsilon$ with probability $9/10$ provided $p = k + \Omega(k)$ and the number of samples satisfies

$$n = \Theta\left(\frac{\sigma^6 + 1}{\varepsilon^2 \lambda_k^6} kd\right).$$

Previously, the same bound[1] was known with a quadratic dependence on $k$ in the case where $p = k$. Here we can strengthen the bound by increasing $p$ slightly.

While we can get some improvements even in the spiked covariance model, our result is substantially more general and applies to any distribution. The sample complexity bound we get varies according to a technical parameter of the distribution. Roughly speaking, we get a near linear sample complexity if the distribution is either "round" (as in the spiked covariance setting) or is very well approximated by a $k$ dimensional subspace. To illustrate the latter condition, we have the following result without making any assumptions other than scaling the distribution:

**Corollary 1.2.** *Let $\mathcal{D}$ be any distribution scaled so that $\Pr\left\{\|z\| > t\right\} \leqslant \exp(-t)$ for every $t \geqslant 1$. Let $U$ represent the top $k$ eigenvectors of the covariance matrix $\mathbb{E} zz^\top$ and $\sigma_1 \geqslant \cdots \geqslant \sigma_d \geqslant 0$ its eigenvalues. Then, SPM invoked with $p = k + \Omega(k)$ outputs a matrix $X_L$ such with probability $9/10$ we have $\left\|(I - X_L X_L^\top)U\right\| \leqslant \varepsilon$ provided SPM receives $n$ samples where $n$ satisfies $n = \tilde{O}\left(\frac{\sigma_k}{\varepsilon^2 k (\sigma_k - \sigma_{k+1})^3} \cdot d\right)$.*

The corollary establishes a sample complexity that's linear in $d$ provided that the spectrum decays quickly, as is common in applications. For example, if the spectrum follows a power law so that $\sigma_j \approx j^{-c}$ for a constant $c > 1/2$, the bound becomes $n = \tilde{O}(k^{2c+2}d/\varepsilon^2)$.

## 1.2 Application to privacy-preserving spectral analysis

Many applications of singular vector computation are plagued by the fact that the underlying matrix contains sensitive information about individuals. A successful paradigm in privacy-preserving data analysis rests on the notion of *differential privacy* which requires all access to the data set to be randomized in such a way that the presence or absence of a single data item is hidden. The notion of data item varies and could either refer to a single entry, a single row, or a rank-1 matrix of bounded norm. More formally, Differential Privacy requires that the output distribution of the algorithm changes only slightly with the addition or deletion of a single data item. This requirement often necessitates the introduction of significant levels of noise that make the computation of various objectives challenging. Differentially private singular vector computation has been studied actively since the work of Blum et al. [BDMN05]. There are two main objectives. The first is computational efficiency. The second objective is to minimize the amount of error that the algorithm introduces.

In this work, we give a fast algorithm for differentially private singular vector computation based on the noisy power method that leads to nearly optimal bounds in a number of settings that were considered in previous work. The algorithm is described in Figure 3. It's a simple instance of NPM in which each noise matrix $G_\ell$ is a gaussian random matrix scaled so that the algorithm achieves $(\varepsilon, \delta)$-differential privacy (as formally defined in Definition E.1). It is easy to see that the algorithm can be implemented in time nearly linear in the number of nonzero entries of the input matrix (input sparsity). This will later lead to strong improvements in running time compared with several previous works.

---

**Input:** Symmetric $A \in \mathbb{R}^{d \times d}$, $L$, $p$, privacy parameters $\varepsilon, \delta > 0$
  1. Let $X_0$ be a random orthonormal basis and put $\sigma = \varepsilon^{-1}\sqrt{4pL\log(1/\delta)}$
  2. For $\ell = 1$ to $L$:
     (a) $Y_\ell \leftarrow AX_{\ell-1} + G_\ell$ where $G_\ell \sim \mathrm{N}(0, \|X_{\ell-1}\|_\infty^2 \sigma^2)^{d \times p}$.
     (b) Compute the QR-factorization $Y_\ell = X_\ell R_\ell$
**Output:** Matrix $X_L$

---

Figure 3: Private Power Method (PPM). Here $\|X\|_\infty = \max_{ij}|X_{ij}|$.

We first state a general purpose analysis of PPM that follows from Corollary 1.1.

**Theorem 1.3.** *Let $k \leqslant p$. Let $U \in \mathbb{R}^{d \times k}$ represent the top $k$ singular vectors of $A$ and let $\sigma_1 \geqslant \cdots \geqslant \sigma_d \geqslant 0$ denote its singular values. Then,* PPM *satisfies $(\varepsilon, \delta)$-differential privacy and after $L = O(\frac{\sigma_k}{\sigma_k - \sigma_{k+1}} \log(d))$ iterations we have with probability $9/10$ that*

$$\left\| (I - X_L X_L^\top)U \right\| \leqslant O\left( \frac{\sigma \max \|X_\ell\|_\infty \sqrt{d \log L}}{\sigma_k - \sigma_{k+1}} \cdot \frac{\sqrt{p}}{\sqrt{p} - \sqrt{k-1}} \right).$$

When $p = k + \Omega(k)$ the trailing factor becomes a constant. If $p = k$ it creates a factor $k$ overhead. In the worst-case we can always bound $\|X_\ell\|_\infty$ by 1 since $X_\ell$ is an orthonormal basis. However, in principle we could hope that a much better bound holds provided that the target subspace $U$ has small coordinates. Hardt and Roth [HR12, HR13] suggested a way to accomplish a stronger bound by considering a notion of *coherence* of $A$, denoted as $\mu(A)$. Informally, the coherence is a well-studied parameter that varies between 1 and $n$, but is often observed to be small. Intuitively, the coherence measures the correlation between the singular vectors of the matrix with the standard basis. Low coherence means that the singular vectors have small coordinates in the standard basis. Many results on matrix completion and robust PCA crucially rely on the assumption that the underlying matrix has low coherence [CR09, CT10, CLMW11] (though the notion of coherence here will be somewhat different).

**Theorem 1.4.** *Under the assumptions of Theorem 1.3, we have the conclusion*

$$\|(I - X_L X_L^\top)U\| \leqslant O\left( \frac{\sigma\sqrt{\mu(A)\log d \log L}}{\sigma_k - \sigma_{k+1}} \cdot \frac{\sqrt{p}}{\sqrt{p} - \sqrt{k-1}} \right) .$$

Hardt and Roth proved this result for the case where $p = 1$. The extension to $p > 1$ lost a factor of $\sqrt{d}$ in general and therefore gave no improvement over Theorem 1.3. Our result resolves the main problem left open in their work. The strength of Theorem 1.4 is that the bound is essentially dimension-free under a natural assumption on the matrix and never worse than our worst-case result. It is also known that in general the dependence on $d$ achieved in Theorem 1.3 is best possible in the worst case (see discussion in [HR13]) so that further progress requires making stronger assumptions. Coherence is a natural such assumption. The proof of Theorem 1.4 proceeds by showing that each iterate $X_\ell$ satisfies $\|X_\ell\|_\infty \leqslant O(\sqrt{\mu(A)\log(d)/d})$ and applying Theorem 1.3. To do this we exploit a non-trivial symmetry of the algorithm that we discuss in Section E.3.

**Other variants of differential privacy.** Our discussion above applied to $(\varepsilon, \delta)$-differential privacy under changing a single entry of the matrix. Several works consider other variants of differential privacy. It is generally easy to adapt the power method to these settings by changing the noise distribution or its scaling. To illustrate this aspect, we consider the problem of privacy-preserving principal component analysis as recently studied by [CSS12, KT13]. Both works consider an algorithm called *exponential mechanism*. The first work gives a heuristic implementation that may not converge, while the second work gives a provably polynomial time algorithm though the running time is more than cubic. Our algorithm gives strong improvements in running time while giving nearly optimal accuracy guarantees as it matches a lower bound of [KT13] up to a $\tilde{O}(\sqrt{k})$ factor. We also improve the error dependence on $k$ by polynomial factors compared to previous work. Moreover, we get an accuracy improvement of $O(\sqrt{d})$ for the case of $(\varepsilon, \delta)$-differential privacy, while these previous works only apply to $(\varepsilon, 0)$-differential privacy. Section E.2 provides formal statements.

## 1.3 Related Work

**Numerical Analysis.** One might expect that a suitable analysis of the noisy power method would have appeared in the numerical analysis literature. However, we are not aware of a reference and there are a number of points to consider. First, our noise model is adaptive thus setting it apart from the classical perturbation theory of the singular vector decomposition [DK70]. Second, we think of the perturbation at each step as large making it conceptually different from floating point errors. Third, research in numerical analysis over the past decades has largely focused on faster Krylov subspace methods. There is some theory of *inexact Krylov methods*, e.g., [SS07] that captures the effect of noisy matrix-vector products in this context. Related to our work are also results on the perturbation stability of the QR-factorization since those could be used to obtain convergence bounds for subspace iteration. Such bounds, however, must depend on the condition number of the matrix that the QR-factorization is applied to. See Chapter 19.9 in [Hig02] and the references therein for background. Our proof strategy avoids this particular dependence on the condition number.

**Streaming PCA.** PCA in the streaming model is related to a host of well-studied problems that we cannot survey completely here. We refer to [ACLS12, MCJ13] for a thorough discussion of prior work. Not mentioned therein is a recent work on incremental PCA [BDF13] that leads to space efficient algorithms computing the top singular vector; however, it's not clear how to extend their results to computing multiple singular vectors.

**Privacy.** There has been much work on differentially private spectral analysis starting with Blum et al. [BDMN05] who used an algorithm known as *Randomized Response* which adds a single noise matrix $N$ either to the input matrix $A$ or the covariance matrix $AA^\top$. This approach appears in a number of papers, e.g. [MM09]. While often easy to analyze it has the disadvantage that it converts sparse matrices to dense matrices and is often impractical on large data sets. Chaudhuri et al. [CSS12] and Kapralov-Talwar [KT13] use the so-called *exponential mechanism* to sample approximate eigenvectors of the matrix. The sampling is done using a heuristic approach without convergence polynomial time convergence guarantees in the first case and using a polynomial time algorithm in the second. Both papers achieve a tight dependence on the matrix dimension $d$ (though

the dependence on $k$ is suboptimal in general). Most closely related to our work are the results of Hardt and Roth [HR13, HR12] that introduced matrix coherence as a way to circumvent existing worst-case lower bounds on the error. They also analyzed a natural noisy variant of power iteration for the case of computing the dominant eigenvector of $A$. When multiple eigenvectors are needed, their algorithm uses the well-known deflation technique. However, this step loses control of the coherence of the original matrix and hence results in suboptimal bounds. In fact, a $\sqrt{\operatorname{rank}(A)}$ factor is lost.

## 1.4 Open Questions

We believe Corollary 1.1 to be a fairly precise characterization of the convergence of the noisy power method to the top $k$ singular vectors when $p = k$. The main flaw is that the noise tolerance depends on the eigengap $\sigma_k - \sigma_{k+1}$, which could be very small. We have some conjectures for results that do not depend on this eigengap.

First, when $p > k$, we think that Corollary 1.1 might hold using the gap $\sigma_k - \sigma_{p+1}$ instead of $\sigma_k - \sigma_{k+1}$. Unfortunately, our proof technique relies on the principal angle decreasing at each step, which does not necessarily hold with the larger level of noise. Nevertheless we expect the principal angle to decrease fairly fast on average, so that $X_L$ will contain a subspace very close to $U$. We are actually unaware of this sort of result even in the noiseless setting.

**Conjecture 1.5.** *Let $X_0$ be a random $p$-dimensional basis for $p > k$. Suppose at every step we have*

$$100\|G_\ell\| \leqslant \varepsilon(\sigma_k - \sigma_{p+1}) \quad and \quad 100\|U^T G_\ell\| \leqslant \frac{\sqrt{p} - \sqrt{k-1}}{\sqrt{d}}$$

*Then with high probability, after $L = O(\frac{\sigma_k}{\sigma_k - \sigma_{p+1}} \log(d/\varepsilon))$ iterations we have*

$$\|(I - X_L X_L^\top)U\| \leqslant \varepsilon.$$

The second way of dealing with a small eigengap would be to relax our goal. Corollary 1.1 is quite stringent in that it requires $X_L$ to approximate the top $k$ singular vectors $U$, which gets harder when the eigengap approaches zero and the $k$th through $p+1$st singular vectors are nearly indistinguishable. A relaxed goal would be for $X_L$ to spectrally approximate $A$, that is

$$\|(I - X_L X_L^\top)A\| \leqslant \sigma_{k+1} + \varepsilon. \tag{1}$$

This weaker goal is known to be achievable in the noiseless setting without any eigengap at all. In particular, [?] shows that (1) happens after $L = O(\frac{\sigma_{k+1}}{\varepsilon} \log n)$ steps in the noiseless setting. A plausible extension to the noisy setting would be:

**Conjecture 1.6.** *Let $X_0$ be a random $2k$-dimensional basis. Suppose at every step we have*

$$\|G_\ell\| \leqslant \varepsilon \quad and \quad \|U^T G_\ell\| \leqslant \varepsilon\sqrt{k/d}$$

*Then with high probability, after $L = O(\frac{\sigma_{k+1}}{\varepsilon} \log d)$ iterations we have that*

$$\|(I - X_L X_L^\top)A\| \leqslant \sigma_{k+1} + O(\varepsilon).$$

## 1.5 Organization

All proofs can be found in the supplementary material. In the remaining space, we limit ourselves to a more detailed discussion of our convergence analysis and the application to streaming PCA. The entire section on privacy is in the supplementary materials in Section E.

## 2 Convergence of the noisy power method

Figure 1 presents our basic algorithm that we analyze in this section. An important tool in our analysis are principal angles, which are useful in analyzing the convergence behavior of numerical eigenvalue methods. Roughly speaking, we will show that the tangent of the $k$-th principal angle between $X$ and the top $k$ eigenvectors of $A$ decreases as $\sigma_{k+1}/\sigma_k$ in each iteration of the noisy power method.

**Definition 2.1** (Principal angles). Let $\mathcal{X}$ and $\mathcal{Y}$ be subspaces of $\mathbb{R}^d$ of dimension at least $k$. The *principal angles* $0 \leqslant \theta_1 \leqslant \cdots \leqslant \theta_k$ between $\mathcal{X}$ and $\mathcal{Y}$ and associated *principal vectors* $x_1, \ldots, x_k$ and $y_1, \ldots, y_k$ are defined recursively via

$$\theta_i(\mathcal{X}, \mathcal{Y}) = \min \left\{ \arccos \left( \frac{\langle x, y \rangle}{\|x\|_2 \|y\|_2} \right) \, : \, x \in \mathcal{X}, y \in \mathcal{Y}, x \perp x_j, y \perp y_j \text{ for all } j < i \right\}$$

and $x_i, y_i$ are the $x$ and $y$ that give this value. For matrices $X$ and $Y$, we use $\theta_k(X, Y)$ to denote the $k$th principal angle between their ranges.

## 2.1   Convergence argument

Fix parameters $1 \leqslant k \leqslant p \leqslant d$. In this section we consider a symmetric $d \times d$ matrix $A$ with singular values $\sigma_1 \geqslant \sigma_2 \geqslant \cdots \geqslant \sigma_d$. We let $U \in \mathbb{R}^{d \times k}$ contain the first $k$ eigenvectors of $A$. Our main lemma shows that $\tan \theta_k(U, X)$ decreases multiplicatively in each step.

**Lemma 2.2.** *Let $U$ contain the largest $k$ eigenvectors of a symmetric matrix $A \in \mathbb{R}^{d \times d}$, and let $X \in \mathbb{R}^{d \times p}$ for $p \geqslant k$. Let $G \in \mathbb{R}^{d \times p}$ satisfy*

$$4\|U^\top G\| \leqslant (\sigma_k - \sigma_{k+1}) \cos \theta_k(U, X)$$
$$4\|G\| \leqslant (\sigma_k - \sigma_{k+1})\varepsilon.$$

*for some $\varepsilon < 1$. Then*

$$\tan \theta_k(U, AX + G) \leqslant \max \left( \varepsilon, \max \left( \varepsilon, \left( \frac{\sigma_{k+1}}{\sigma_k} \right)^{1/4} \right) \tan \theta_k(U, X) \right).$$

We can inductively apply the previous lemma to get the following general convergence result.

**Theorem 2.3.** *Let $U$ represent the top $k$ eigenvectors of the matrix $A$ and $\gamma = 1 - \sigma_{k+1}/\sigma_k$. Suppose that the initial subspace $X_0$ and noise $G_\ell$ is such that*

$$5\|U^\top G_\ell\| \leqslant (\sigma_k - \sigma_{k+1}) \cos \theta_k(U, X_0)$$
$$5\|G_\ell\| \leqslant \varepsilon(\sigma_k - \sigma_{k+1})$$

*at every stage $\ell$, for some $\varepsilon < 1/2$. Then there exists an $L \lesssim \frac{1}{\gamma} \log \left( \frac{\tan \theta_k(U, X_0)}{\varepsilon} \right)$ such that for all $\ell \geqslant L$ we have $\tan \theta(U, X_L) \leqslant \varepsilon$.*

## 2.2   Random initialization

The next lemma essentially follows from bounds on the smallest singular value of gaussian random matrices [RV09].

**Lemma 2.4.** *For an arbitrary orthonormal $U$ and random subspace $X$, we have*

$$\tan \theta_k(U, X) \leqslant \tau \frac{\sqrt{d}}{\sqrt{p} - \sqrt{k-1}}$$

*with all but $\tau^{-\Omega(p+1-k)} + e^{-\Omega(d)}$ probability.*

With this lemma we can prove the corollary that we stated in the introduction.

*Proof of Corollary 1.1.* By Lemma 2.4, with the desired probability we have $\tan \theta_k(U, X_0) \leqslant \frac{\tau\sqrt{d}}{\sqrt{p} - \sqrt{k-1}}$. Hence $\cos \theta_k(U, X_0) \geqslant 1/(1 + \tan \theta_k(U, X_0)) \geqslant \frac{\sqrt{p} - \sqrt{k-1}}{2 \cdot \tau\sqrt{d}}$. Rescale $\tau$ and apply Theorem 2.3 to get that $\tan \theta_k(U, X_L) \leqslant \varepsilon$. Then $\|(I - X_L X_L^\top)U\| = \sin \theta_k(U, X_L) \leqslant \tan \theta_k(U, X_L) \leqslant \varepsilon$. ∎

# 3 Memory efficient streaming PCA

In the streaming PCA setting we receive a stream of samples $z_1, z_2, \cdots \in \mathbb{R}^d$. Each sample is drawn i.i.d. from an unknown distribution $\mathcal{D}$ over $\mathbb{R}^d$. Our goal is to compute the dominant $k$ eigenvectors of the covariance matrix $A = \mathbb{E}_{z \sim \mathcal{D}} zz^\top$. The challenge is to do this with small space, so we cannot store the $d^2$ entries of the sample covariance matrix. We would like to use $O(dk)$ space, which is necessary even to store the output.

The streaming power method (Figure 2, introduced by [MCJ13]) is a natural algorithm that performs streaming PCA with $O(dk)$ space. The question that arises is how many samples it requires to achieve a given level of accuracy, for various distributions $\mathcal{D}$. Using our general analysis of the noisy power method, we show that the streaming power method requires fewer samples and applies to more distributions than was previously known. We analyze a broad class of distributions:

**Definition 3.1.** A distribution $\mathcal{D}$ over $\mathbb{R}^d$ is $(B, p)$-*round* if for every $p$-dimensional projection $P$ and all $t \geqslant 1$ we have $\mathrm{Pr}_{z \sim \mathcal{D}} \{\|z\| > t\} \leqslant \exp(-t)$ and $\mathrm{Pr}_{z \sim \mathcal{D}} \left\{ \|Pz\| > t \cdot \sqrt{Bp/d} \right\} \leqslant \exp(-t)$.

The first condition just corresponds to a normalization of the samples drawn from $\mathcal{D}$. Assuming the first condition holds, the second condition always holds with $B = d/p$. For this reason our analysis in principle applies to any distribution, but the sample complexity will depend quadratically on $B$.

Let us illustrate this definition through the example of the spiked covariance model studied by [MCJ13]. The spiked covariance model is defined by an orthonormal basis $U \in \mathbb{R}^{d \times k}$ and a diagonal matrix $\Lambda \in \mathbb{R}^{k \times k}$ with diagonal entries $\lambda_1 \geqslant \lambda_2 \geqslant \cdots \geqslant \lambda_k > 0$. The distribution $\mathcal{D}(U, \Lambda)$ is defined as the normal distribution $\mathrm{N}(0, (U\Lambda^2 U^\top + \sigma^2 \mathrm{Id}_{d \times d})/D)$ where $D = \Theta(d\sigma^2 + \sum_i \lambda_i^2)$ is a normalization factor chosen so that the distribution satisfies the norm bound. Note that the the $i$-th eigenvalue of the covariance matrix is $\sigma_i = (\lambda_i^2 + \sigma^2)/D$ for $1 \leqslant i \leqslant k$ and $\sigma_i = \sigma^2/D$ for $i > k$. We show in Lemma D.2 that the spiked covariance model $\mathcal{D}(U, \Lambda)$ is indeed $(B, p)$-round for $B = O(\frac{\lambda_1^2 + \sigma^2}{\mathrm{tr}(\Lambda)/d + \sigma^2})$, which is constant for $\sigma \gtrsim \lambda_1$. We have the following main theorem.

**Theorem 3.2.** *Let $\mathcal{D}$ be a $(B, p)$-round distribution over $\mathbb{R}^d$ with covariance matrix $A$ whose eigenvalues are $\sigma_1 \geqslant \sigma_2 \geqslant \cdots \geqslant \sigma_d \geqslant 0$. Let $U \in \mathbb{R}^{d \times k}$ be an orthonormal basis for the eigenvectors corresponding to the first $k$ eigenvalues of $A$. Then, the streaming power method* SPM *returns an orthonormal basis $X \in \mathbb{R}^{d \times p}$ such that $\tan \theta(U, X) \leqslant \varepsilon$ with probability $9/10$ provided that* SPM *receives $n$ samples from $\mathcal{D}$ for some $n$ satisfying*

$$n \leqslant \tilde{O} \left( \frac{B^2 \sigma_k k \log^2 d}{\varepsilon^2 (\sigma_k - \sigma_{k+1})^3 d} \right)$$

*if $p = k + \Theta(k)$. More generally, for all $p \geqslant k$ one can get the slightly stronger result*

$$n \leqslant \tilde{O} \left( \frac{B p \sigma_k \max\{1/\varepsilon^2, Bp/(\sqrt{p} - \sqrt{k-1})^2\} \log^2 d}{(\sigma_k - \sigma_{k+1})^3 d} \right).$$

Instantiating with the spiked covariance model gives the following:

**Corollary 3.3.** *In the spiked covariance model $\mathcal{D}(U, \Lambda)$ the conclusion of Theorem 3.2 holds for $p = 2k$ with*

$$n = \tilde{O} \left( \frac{(\lambda_1^2 + \sigma^2)^2 (\lambda_k^2 + \sigma^2)}{\varepsilon^2 \lambda_k^6} dk \right).$$

When $\lambda_1 = O(1)$ and $\lambda_k = \Omega(1)$ this becomes $n = \tilde{O} \left( \frac{\sigma^6 + 1}{\varepsilon^2} \cdot dk \right)$.

We can apply Theorem 3.2 to all distributions that have exponentially concentrated norm by setting $B = d/p$. This gives the following result.

**Corollary 3.4.** *Let $\mathcal{D}$ be any distribution scaled such that $\mathrm{Pr}_{z \sim \mathcal{D}}[\|z\| > t] \leqslant \exp(-t)$ for all $t \geqslant 1$. Then the conclusion of Theorem 3.2 holds for $p = 2k$ with*

$$n = \tilde{O} \left( \frac{\sigma_k}{\varepsilon^2 k (\sigma_k - \sigma_{k+1})^3} \cdot d \right).$$

If the eigenvalues follow a power law, $\sigma_j \approx j^{-c}$ for a constant $c > 1/2$, this gives an $n = \tilde{O}(k^{2c+2} d/\varepsilon^2)$ bound on the sample complexity.

## Footnotes

[1] That the bound stated in [MCJ13] has a $\sigma^6$ dependence is not completely obvious. There is a $O(\sigma^4)$ in the numerator and $\log((\sigma^2 + 0.75\lambda_k^2)/(\sigma^2 + 0.5\lambda_k^2))$ in the denominator which simplifies to $O(1/\sigma^2)$ for constant $\lambda_k$ and $\sigma^2 \geqslant 1$.

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
