[Supplementary Material]

# A    Deferred Concentration Inequalities

**Theorem A.1** (Chernoff bound). *Let the random variables $X_1, \ldots, X_m$ be independent random variables such that for every $i$, $X_i \in [-1, 1]$ almost surely. Let $X = \sum_{i=1}^{m} X_i$ and let $\sigma^2 = \mathbb{V} X$. Then, for any $t > 0$,*
$$\Pr\{|X - \mathbb{E} X| > t\} \leqslant \exp\left(-\frac{t^2}{4\sigma^2}\right) .$$

The next lemma follows from standard concentration properties of the Gaussian distribution.

**Lemma A.2.** *Let $U \in \mathbb{R}^{d \times k}$ be a matrix with orthonormal columns. Let $G_1, \ldots, G_L \sim N(0, \sigma^2)^{d \times p}$ with $k \leqslant p \leqslant d$ and assume that $L \leqslant d$. Then, with probability $1 - 10^{-4}$,*
$$\max_{\ell \in [L]} \|U^\top G_\ell\| \leqslant O\left(\sigma\sqrt{p + \log L}\right) .$$

*Proof.* By rotational invariance of $G_\ell$ the spectral norm $\|U^\top G_\ell\|$ is distributed like largest singular value of a random draw from $k \times p$ gaussian matrix $N(0, \sigma^2)^{k \times p}$. Since $p \geqslant k$, the largest singular value strongly concentrates around $O(\sigma\sqrt{p})$ with a gaussian tail. By the gaussian concentration of Lipschitz functions of gaussians, taking the maximum over $L$ gaussian matrices introduces an additive $O(\sigma\sqrt{\log L})$ term. ∎

We also have an analogue of the previous lemma for the Laplacian distribution.

**Lemma A.3.** *Let $U \in \mathbb{R}^{n \times k}$ be a matrix with orthonormal columns. Let $G_1, \ldots, G_L \sim \mathrm{Lap}(0, \lambda)^{d \times p}$ with $k \leqslant p \leqslant d$ and assume that $L \leqslant d$. Then, with probability $1 - 10^{-4}$,*
$$\max_{\ell \in [L]} \|U^\top G_\ell\| \leqslant O\left(\lambda\sqrt{pk}\log(Lpk)\right) .$$

*Proof.* We claim that with probability $1 - 10^{-4}$ for every $\ell \in [L]$, every entry of $U^\top G_\ell$ is bounded by $O(\lambda \log(Lpk))$ in absolute value. This follows because each entry has variance $\lambda^2$ and is a weighted sum of $n$ independent Laplacian random variables $\mathrm{Lap}(0, \lambda)$. Assuming this event occurs, we have
$$\max_{\ell \in [L]} \|U^\top G_\ell\| \leqslant \max_{\ell \in [L]} \|U^\top G_\ell\|_F \leqslant O\left(\lambda\sqrt{pk}\log(Lpk)\right) . \quad ∎$$

**Lemma A.4** (Matrix Chernoff). *Let $X_1, \ldots, X_n \sim \mathcal{X}$ be i.i.d. random matrices of maximum dimension $d$ and mean $\mu$, uniformly bounded by $\|X\| \leqslant R$. Then for all $t \leqslant 1$,*
$$\Pr\left\{\left\|\tfrac{1}{n}\sum_i X_i - \mathbb{E} X_1\right\| \geqslant tR\right\} \leqslant de^{-\Omega(mt^2)}$$

# B    Reduction to symmetric matrices

For all our purposes it suffices to consider symmetric $n \times n$ matrices. Given a non-symmetric $m \times n$ matrix $B$ we may always consider the $(m + n) \times (m + n)$ matrix $A = [\,0\,B \,|\, B^\top\,0\,]$. This transformation preserves all the parameters that we are interested in as was argued in [HR13] more formally. This allows us to discuss symmetric eigendecompositions rather than singular vector decompositions and therefore simplify our presentation below.

# C    Proof for Convergence of Noisy Power Method

We will make use of a non-recursive expression for the principal angles, defined in terms of the set $\mathcal{P}_k$ of $p \times p$ projection matrices $\Pi$ from $p$ dimensions to $k$ dimensional subspaces:

**Claim C.1.** *Let $U \in \mathbb{R}^{d \times k}$ have orthonormal columns and $X \in \mathbb{R}^{d \times p}$ have independent columns, for $p \geqslant k$. Then*
$$\cos\theta_k(U, X) = \min_{\Pi \in \mathcal{P}_k} \min_{\substack{x \in \mathrm{range}(X\Pi) \\ \|x\|_2 = 1}} \|U^\top x\| = \min_{\Pi \in \mathcal{P}_k} \min_{\substack{\|w\|_2 = 1 \\ \Pi w = w}} \frac{\|U^\top X w\|}{\|X w\|}.$$

*For $V = U^\perp$, we have*
$$\tan\theta_k(U, X) = \min_{\Pi \in \mathcal{P}_k} \max_{x \in \mathrm{range}(X\Pi)} \frac{\|V^\top x\|}{\|U^\top x\|} = \min_{\Pi \in \mathcal{P}_k} \max_{\substack{\|w\|_2 = 1 \\ \Pi w = w}} \frac{\|V^\top X w\|}{\|U^\top X w\|}.$$

*Proof of Lemma 2.2.* Let $\Pi^*$ be the matrix projecting onto the smallest $k$ principal angles of $X$, so that

$$\tan\theta_k(U, X) = \max_{\substack{\|w\|_2=1 \\ \Pi^* w=w}} \frac{\|V^\top Xw\|}{\|U^\top Xw\|}.$$

We have that

$$\begin{aligned}
\tan\theta_k(U, AX+G) &= \min_{\Pi\in\mathcal{P}_k} \max_{\substack{\|w\|_2=1 \\ \Pi w=w}} \frac{\|V^\top(AX+G)w\|}{\|U^\top(AX+G)w\|} \\
&\leqslant \max_{\substack{\|w\|_2=1 \\ \Pi^* w=w}} \frac{\|V^\top AXw\| + \|V^\top Gw\|}{\|U^\top AXw\| - \|U^\top Gw\|} \\
&\leqslant \max_{\substack{\|w\|_2=1 \\ \Pi^* w=w}} \frac{1}{\|U^\top Xw\|} \cdot \frac{\sigma_{k+1}\|V^\top Xw\| + \|V^\top Gw\|}{\sigma_k - \|U^\top Gw\|/\|U^\top Xw\|}
\end{aligned} \tag{2}$$

Define $\Delta = (\sigma_k - \sigma_{k+1})/4$. By the assumption on $G$,

$$\max_{\substack{\|w\|_2=1 \\ \Pi^* w=w}} \frac{\|U^\top Gw\|}{\|U^\top Xw\|} \leqslant \|U^\top G\|/\cos\theta_k(U, X) \leqslant (\sigma_k - \sigma_{k+1})/4 = \Delta.$$

Similarly, and using that $1/\cos\theta \leqslant 1 + \tan\theta$ for any angle $\theta$,

$$\max_{\substack{\|w\|_2=1 \\ \Pi^* w=w}} \frac{\|V^\top Gw\|}{\|U^\top Xw\|} \leqslant \|G\|/\cos\theta_k(U, X) \leqslant \varepsilon\Delta(1 + \tan\theta_k(U, X)).$$

Plugging back into (2) and using $\sigma_k = \sigma_{k+1} + 4\Delta$,

$$\begin{aligned}
\tan\theta_k(U, AX+G) &\leqslant \max_{\substack{\|w\|_2=1 \\ \Pi^* w=w}} \frac{\|V^\top Xw\|}{\|U^\top Xw\|} \cdot \frac{\sigma_{k+1}}{\sigma_{k+1}+3\Delta} + \frac{\varepsilon\Delta(1+\tan\theta_k(U, X))}{\sigma_{k+1}+3\Delta}. \\
&= \frac{\sigma_{k+1}+\varepsilon\Delta}{\sigma_{k+1}+3\Delta}\tan\theta_k(U, X) + \frac{\varepsilon\Delta}{\sigma_{k+1}+3\Delta} \\
&= (1 - \frac{\Delta}{\sigma_{k+1}+3\Delta})\frac{\sigma_{k+1}+\varepsilon\Delta}{\sigma_{k+1}+2\Delta}\tan\theta_k(U, X) + \frac{\Delta}{\sigma_{k+1}+3\Delta}\varepsilon \\
&\leqslant \max(\varepsilon, \frac{\sigma_{k+1}+\varepsilon\Delta}{\sigma_{k+1}+2\Delta}\tan\theta_k(U, X))
\end{aligned}$$

where the last inequality uses that the weighted mean of two terms is less than their maximum. Finally, we have that

$$\frac{\sigma_{k+1}+\varepsilon\Delta}{\sigma_{k+1}+2\Delta} \leqslant \max(\frac{\sigma_{k+1}}{\sigma_{k+1}+\Delta}, \varepsilon)$$

because the left hand side is a weighted mean of the components on the right. Since $\frac{\sigma_{k+1}}{\sigma_{k+1}+\Delta} \leqslant (\frac{\sigma_{k+1}}{\sigma_{k+1}+4\Delta})^{1/4} = (\sigma_{k+1}/\sigma_k)^{1/4}$, this gives the result. ∎

*Proof of Theorem 2.3.* We will see that at every stage $\ell$ of the algorithm,

$$\tan\theta_k(U, X_\ell) \leqslant \max(\varepsilon, \tan\theta_k(U, X_0))$$

which implies for $\varepsilon \leqslant 1/2$ that

$$\cos\theta_k(U, X_\ell) \geqslant \min(1 - \varepsilon^2/2, \cos\theta_k(U, X_0)) \geqslant \frac{7}{8}\cos\theta_k(U, X_0)$$

so Lemma 2.2 applies at every stage. This means that

$$\tan\theta_k(U, X_{\ell+1}) = \tan\theta_k(U, AX_\ell + G) \leqslant \max(\varepsilon, \delta\tan\theta_k(U, X_\ell))$$

for $\delta = \max(\varepsilon, (\sigma_{k+1}/\sigma_k)^{1/4})$. After

$$L = \log_{1/\delta} \frac{\tan \theta_k(U, X_0)}{\varepsilon}$$

iterations the tangent will reach $\varepsilon$ and remain there. Observing that

$$\log(1/\delta) \gtrsim \min(\log(1/\varepsilon), \log(\sigma_k/\sigma_{k+1})) \geqslant \min(1, \log \frac{1}{1-\gamma}) \geqslant \min(1, \gamma) = \gamma$$

gives the result. ∎

## C.1 Random initialization

*Proof of Lemma 2.4.* Consider the singular value decomposition $U^\top X = A\Sigma B^\top$ of $U^\top X$. Setting $\Pi$ to be matrix projecting onto the first $k$ columns of $B$, we have that

$$\tan \theta_k(U, X) \leqslant \max_{\substack{\|w\|_2=1 \\ \Pi w = w}} \frac{\|V^\top X w\|}{\|U^\top X w\|} \leqslant \|V^\top X\| \max_{\substack{\|w\|_2=1 \\ \Pi w = w}} \frac{1}{\|\Sigma B^\top w\|} = \|V^\top X\| \max_{\substack{\|w\|_2=1 \\ \mathrm{supp}(w) \in [k]}} \frac{1}{\|\Sigma w\|} = \frac{\|V^\top X\|}{\sigma_k(U^\top X)}.$$

Let $X \sim N(0, I_{d\times p})$ represent the random subspace. Then $Y := U^\top X \sim N(0, I_{k\times p})$. By [RV09], for any $\varepsilon$, the smallest singular value of $Y$ is at least $(\sqrt{p} - \sqrt{k-1})/\tau$ with all but $\tau^{-\Omega(p+1-k)} + e^{-\Omega(p)}$ probability. On the other hand, $\|X\| \lesssim \sqrt{d}$ with all but $e^{-\Omega(d)}$ probability. Hence

$$\tan \theta_k(U, X) \lesssim \tau \frac{\sqrt{d}}{\sqrt{p} - \sqrt{k-1}}$$

with the desired probability. Rescaling $\tau$ gets the result. ∎

# D   Error term analysis for Streaming PCA

Fix an orthonormal basis $X \in \mathbb{R}^{d\times k}$. Let $z_1, \ldots, z_n \sim \mathcal{D}$ be samples from a distribution $\mathcal{D}$ with covariance matrix $A$ and consider the matrix

$$G = \left(A - \widehat{A}\right) X,$$

where $\widehat{A} = \frac{1}{n} \sum_{i=1}^n z_i z_i^\top$ is the empirical covariance matrix on $n$ samples. Then, we have that $\widehat{A}X = AX + G$. In other words, one update step of the power method executed on $\widehat{A}$ can be expressed as an update step on $A$ with noise matrix $G$. This simple observation allows us to apply our analysis of the noisy power method to this setting after obtaining suitable bounds on $\|G\|$ and $\|U^\top G\|$.

**Lemma D.1.** *Let $\mathcal{D}$ be a $(B, p)$-round distribution with covariance matrix $M$. Then with all but $O(1/n^2)$ probability,*

$$\|G\| \lesssim \sqrt{\frac{Bp \log^4 n \log d}{dn}} + \frac{1}{n^2} \quad and \quad \|U^\top G\| \lesssim \sqrt{\frac{B^2 p^2 \log^4 n \log d}{d^2 n}} + \frac{1}{n^2}$$

*Proof.* We will use a matrix Chernoff bound to show that

1. $\Pr\left\{ \|G\| > Ct \log(n)^2 \sqrt{Bp/d} + O(1/n^2) \right\} \leqslant d \exp(-t^2 n) + 1/n^2$

2. $\Pr\left\{ \|U^\top G\| > Ct \log(n)^2 Bp/d + O(1/n^2) \right\} \leqslant d \exp(-t^2 n) + 1/n^2$

setting $t = \sqrt{\frac{2}{n} \log d}$ gives the result. However, matrix Chernoff inequality requires the distribution to satisfy a norm bound with probability 1. We will therefore create a closely related distribution $\tilde{\mathcal{D}}$ that satisfies such a norm

constraint and is statistically indistinguishable up to small error on $n$ samples. We can then work with $\tilde{\mathcal{D}}$ instead of $\mathcal{D}$. This truncation step is standard and works because of the concentration properties of $\mathcal{D}$.

Indeed, let $\tilde{\mathcal{D}}$ be the distribution obtained from $\mathcal{D}$ be replacing a sample $z$ with $0$ if

$$\|z\| > C \log(n) \quad \text{or} \quad \|U^\top z\| \geqslant C \log(n)\sqrt{Bp/d} \quad \text{or} \quad \|z^\top X\| > C \log(n)\sqrt{Bp/d} \,.$$

For sufficiently large constant $C$, it follows from the definition of $(B, p)$-round that the probability that one or more of $n$ samples from $\mathcal{D}$ get zeroed out is at most $1/n^2$. In particular, the two product distributions $\mathcal{D}^{(n)}$ and $\tilde{\mathcal{D}}^{(n)}$ have total variation distance at most $1/n^2$. Furthermore, we claim that the covariance matrices of the two distributions are at most $O(1/n^2)$ apart in spectral norm. Formally,

$$\left\| \mathop{\mathbb{E}}_{z \sim \mathcal{D}} zz^\top - \mathop{\mathbb{E}}_{\tilde{z} \sim \tilde{\mathcal{D}}} \tilde{z}\tilde{z}^\top \right\| \leqslant \frac{1}{n^2} \cdot O\left( \int_{t \geqslant 1} C^2 t^2 \log^2(n) \exp(-t) \mathrm{d}t \right) \leqslant O(1/n^2) \,.$$

In the first inequality we use the fact that $z$ only gets zeroed out with probability $1/n^2$. Conditional on this event, the norm of $z$ is larger than $tC \log(n)$ with probability at most $n^2 \exp(-\frac{1}{2}tC \log n) \leqslant \exp(-t)$. Assuming the norm is at most $tC \log(n)$ we have $\|zz^\top\| \leqslant t^2 C^2 \log^2(n)$ and this bounds the contribution to the spectral norm of the difference.

Now let $\tilde{G}$ be the error matrix defined as $G$ except that we replace the samples $z_1, \ldots, z_n$ by $n$ samples $\tilde{z}_1, \ldots, \tilde{z}_n$ from the truncated distribution $\tilde{\mathcal{D}}$. By our preceding discussion, it now suffices to show that

1. $\Pr\left\{ \|\tilde{G}\| > Ct \log^2(n)\sqrt{Bp/d} \right\} \leqslant d \exp(-t^2 n)$

2. $\Pr\left\{ \|U^\top \tilde{G}\| > Ct \log^2(n)Bp/d \right\} \leqslant d \exp(-t^2 n)$

To see this, let $S_i = \tilde{z}_i \tilde{z}_i^\top X$. We have

$$\|S_i\| \leqslant \|\tilde{z}_i\| \cdot \|\tilde{z}_i^\top X\| \leqslant C^2 \log^2(n) \cdot \sqrt{Bp/d}$$

Similarly,

$$\|U^\top S_i\| \leqslant \|U^\top \tilde{z}_i\| \cdot \|\tilde{z}_i^\top X\| \leqslant C^2 \log^2(n) \cdot \frac{Bp}{d} \,.$$

The claims now follow directly from the matrix Chernoff bound stated in Lemma A.4. ∎

## D.1 Proof of Theorem 3.2

Given Lemma D.1 we will choose $n$ such that the error term in each iteration satisfies the assumptions of Theorem 2.3. Let $G_\ell$ denote the instance of the error term $G$ arising in the $\ell$-th iteration of the algorithm. We can find an $n$ satisfying

$$\frac{n}{\log(n)^4} = O\left( \frac{Bp \max\left\{ 1/\varepsilon^2, Bp/(\sqrt{p} - \sqrt{k-1})^2 \right\} \log d}{(\sigma_k - \sigma_{k+1})^2 d} \right)$$

such that by Lemma D.1 we have that with probability $1 - O(1/n^2)$,

$$\|G_\ell\| \leqslant \frac{\varepsilon(\sigma_k - \sigma_{k+1})}{5} \quad \text{and} \quad \|U^\top G_\ell\| \leqslant \frac{\sigma_k - \sigma_{k+1}}{5} \frac{\sqrt{p} - \sqrt{k-1}}{\sqrt{d}} \,.$$

Here we used that by definition $1/n \ll \varepsilon$ and $1/n \ll \sigma_k - \sigma_{k+1}$ and so the $1/n^2$ term in Lemma D.1 is of lower order.

With this bound, it follows from Theorem 2.3 that after $L = O(\log(d/\varepsilon)/(1 - \sigma_{k+1}/\sigma_k))$ iterations we have with probability $1 - \max\{1, L/n^2\}$ that $\tan \theta(U, X_L) \leqslant \varepsilon$. The over all sample complexity is therefore

$$Ln = \tilde{O}\left( \frac{Bp\sigma_k \max\left\{ 1/\varepsilon^2, Bp/(\sqrt{p} - \sqrt{k-1})^2 \right\} \log^2 d}{(\sigma_k - \sigma_{k+1})^3 d} \right) \,.$$

Here we used that $1 - \sigma_{k+1}/\sigma_k = (\sigma_k - \sigma_{k+1})/\sigma_k$. This concludes the proof of Theorem 3.2.

### D.2 Proof of Lemma D.2 and Corollary 3.4

**Lemma D.2.** *The spiked covariance model $\mathcal{D}(U, \Lambda)$ is $(B, k)$-round for $B = O(\frac{\lambda_1^2 + \sigma^2}{\operatorname{tr}(\Lambda)/d + \sigma^2})$.*

*Proof.* Note that an example $z \sim \mathcal{D}(U, \Lambda)$ is distributed as $U\Lambda g + g'$ where $g \sim \mathrm{N}(0, 1/D)^k$ is a standard gaussian and $g' \sim \mathrm{N}(0, \sigma^2/D)^d$. is a noise term. Recall, that $D$ is the normalization term. Let $P$ be any projection operator onto a $k$-dimensional space. Then,

$$\|Pz\| = \|PU\Lambda g + Pg'\| \leqslant \|PU\Lambda g\| + \|Pg'\| \leqslant \|\Lambda g\| + \|Pg'\| \leqslant \lambda_1 \|g\| + \|Pg'\|.$$

By rotational invariance of $g'$, we may assume that $P$ is the projection onto the first $k$ coordinates. Hence, $\|Pg'\|$ is distributed like the norm of $\mathrm{N}(0, \sigma^2/D)^k$. Using standard tail bounds for the norm of a gaussian random variables, we can see that $\|Pz\|^2 = O(t(k\lambda_1^2 + k\sigma^2)/D)$ with probability $1 - \exp(-t)$. On the other hand, $D = \Theta(\sum_{i=1}^{k} \lambda_i^2 + d\sigma^2)$. We can now solve for $B$ by setting

$$\Theta(\frac{k\lambda_1^2 + k\sigma^2}{\sum_{i=1}^{k} \lambda_i^2 + d\sigma^2}) = \frac{Bk}{d} \quad \Leftrightarrow \quad B = \Theta(\frac{\lambda_1^2 + \sigma^2}{\frac{1}{d}\sum_{i=1}^{k} \lambda_i^2 + \sigma^2}).$$

∎

Corollary 3.4 follows by plugging in the bound on $B$ and the eigenvalues of the covariance matrix into our main theorem.

*Proof of Corollary 3.4.* In the spiked covariance model $\mathcal{D}(U, \Lambda)$ we have

$$B = \frac{\lambda_1^2 + \sigma^2}{D}, \quad \sigma_k = \frac{\lambda_k^2 + \sigma^2}{D}, \quad \sigma_{k+1} = \frac{\sigma^2}{D}, \quad D = O(\operatorname{tr}(\Lambda^2) + d\sigma^2).$$

Hence,

$$\frac{B^2 \sigma_k}{(\sigma_k - \sigma_{k+1})^3 d} = \frac{(\lambda_1^2 + \sigma^2)^2(\lambda_k^2 + \sigma^2)}{\lambda_k^6 d} \leqslant \frac{(\lambda_1^2 + \sigma^2)^3}{\lambda_k^6 d}$$

Plugging this bound into Theorem 3.2 gives Corollary 3.4. ∎

## E  Privacy-preserving singular vector computation

In this section we prove our results about privacy-preserving singular vector computation. We begin with a standard definition of differential privacy, sometimes referred to as *entry-level differential privacy*, as it hides the presence or absence of a single entry.

**Definition E.1** (Differential Privacy). A randomized algorithm $M \colon \mathbb{R}^{d \times d'} \to R$ (where $R$ is some arbitrary abstract range) is $(\varepsilon, \delta)$-*differentially private* if for all pairs of matrices $A, A' \in \mathbb{R}^{d \times d'}$ differing in only one entry by at most 1 in absolute value, we have that for all subsets of the range $S \subseteq R$, the algorithm satisfies: $\Pr\{M(A) \in S\} \leqslant \exp(\varepsilon) \Pr\{M(A') \in S\} + \delta$.

The definition is most meaningful when $A$ has entries in $[0, 1]$ so that the above definition allows for a single entry to change arbitrarily within this range. However, this is not a requirement for us. The privacy guarantee can be strengthened by decreasing $\varepsilon > 0$.

For our choice of $\sigma$ in Figure 3 the algorithm satisfies $(\varepsilon, \delta)$-differential privacy as follows easily from properties of the Gaussian distribution. See, for example, [HR13] for a proof.

**Claim E.2.** PPM *satisfies $(\varepsilon, \delta)$-differential privacy.*

It is straightforward to prove Theorem 1.3 by invoking our convergence analysis of the noisy power method together with suitable error bounds. The error bounds are readily available as the noise term is just gaussian.

*Proof of Theorem 1.3.* Let $m = \max \|X_\ell\|_\infty$. By Lemma A.2 the following bounds hold with probability $99/100$:

1. $\max_{\ell=1}^{L} \|G_\ell\| \lesssim \sigma m \sqrt{d \log L}$

2. $\max_{\ell=1}^{L} \|U^\top G_\ell\| \lesssim \sigma m \sqrt{k \log L}$

Let

$$\varepsilon' = \frac{\sigma m \sqrt{d \log L}}{\sigma_k - \sigma_{k+1}} \gtrsim \frac{5 \max_{\ell=1}^{L} \|G_\ell\|}{\sigma_k - \sigma_{k+1}}.$$

By Corollary 1.1, if we also have that $\max_{\ell=1}^{L} \|U^\top G_\ell\| \leqslant (\sigma_k - \sigma_{k+1}) \frac{\sqrt{p} - \sqrt{k-1}}{\tau \sqrt{d}}$ for a sufficiently large constant $\tau$, then we will have that

$$\|(I - X_L X_L^\top) U\| \leqslant \varepsilon' \leqslant \frac{\sigma m \sqrt{d \log L}}{\sigma_k - \sigma_{k+1}}$$

after the desired number of iterations, giving the theorem. Otherwise,

$$(\sigma_k - \sigma_{k+1}) \frac{\sqrt{p} - \sqrt{k-1}}{\tau \sqrt{d}} \leqslant \max_{\ell=1}^{L} \|U^\top G_\ell\| \lesssim \varepsilon' (\sigma_k - \sigma_{k+1}) \sqrt{k/d},$$

so it is trivially true that

$$\frac{\sigma m \sqrt{d \log L}}{\sigma_k - \sigma_{k+1}} \frac{\sqrt{p}}{\sqrt{p} - \sqrt{k-1}} \geqslant \varepsilon' \frac{\sqrt{k}}{\sqrt{p} - \sqrt{k-1}} \gtrsim 1 \geqslant \|(I - X_L X_L^\top) U\|.$$

$\blacksquare$

## E.1 Low-rank approximation

Our results readily imply that we can compute accurate differentially private low-rank approximations. The main observation is that, assuming $X_L$ and $U$ have the same dimension, $\tan \theta(U, X_L) \leqslant \alpha$ implies that the matrix $X_L$ also leads to a good low-rank approximation for $A$ in the spectral norm. In particular

$$\|(I - X_L X_L^\top) A\| \leqslant \sigma_{k+1} + \alpha \sigma_1. \tag{3}$$

Moreover the projection step of computing $X_L X_L^\top A$ can be carried out easily in a privacy-preserving manner. It is again the $\ell_\infty$-norm of the columns of $X_L$ that determine the magnitude of noise that is needed. Since $A$ is symmetric, we have $X^\top A = (AX)^\top$. Hence, to obtain a good low-rank approximation it suffices to compute the product $AX_L$ privately as $AX_L + G_L$. This leads to the following corollary.

**Corollary E.3.** *Let $A \in \mathbb{R}^{d \times d}$ be a symmetric matrix with singular values $\sigma_1 \geqslant \ldots \geqslant \sigma_d$ and let $\gamma = 1 - \sigma_{k+1}/\sigma_k$. There is an $(\varepsilon, \delta)$-differentially private algorithm that given $A$ and $k$, outputs a rank $2k$ matrix $B$ such that with probability $9/10$,*

$$\|A - B\| \leqslant \sigma_{k+1} + \tilde{O}\left( \frac{\sigma_1 \sqrt{(k/\gamma) d \log d \log(1/\delta)}}{\varepsilon(\sigma_k - \sigma_{k+1})} \right).$$

*The $\tilde{O}$-notation hides the factor $O\left( \sqrt{\log(\log(d)/\gamma)} \right)$.*

*Proof.* Apply Theorem 1.3 with $p = 2k$ and run the algorithm for $L + 1$ steps with $L = O(\gamma^{-1} \log d)$. This gives the bound

$$\alpha = \|(I - X_L X_L^\top) A\| \leqslant O\left( \frac{\sqrt{(k/\gamma) d \log d \log(\log(d)/\gamma) \log(1/\delta)}}{\varepsilon(\sigma_k - \sigma_{k+1})} \right).$$

Moreover, the algorithm has computed $Y_{L+1} = AX_L + G_L$ and we have $B = X_L Y_{L+1}^\top = X_L X_L^\top A + X_L G_L^\top$. Therefore

$$\|A - B\| \leqslant \sigma_{k+1} + \alpha \sigma_1 + \|X_L G_L^\top\|$$

where $\|X_L G_L^\top\| \leqslant \|G_L\|$. By definition of the algorithm and Lemma A.2, we have

$$\|G_L\| \leqslant O\left( \sqrt{\sigma^2 d} \right) = O\left( \frac{1}{\varepsilon} \sqrt{(k/\gamma) d \log(d) \log(1/\delta)} \right).$$

Given that the $\alpha$-term gets multiplied by $\sigma_1$, this bound on $\|G_L\|$ is of lower order and the corollary follows. $\blacksquare$

## E.2 Principal Component Analysis

Here we illustrate that our bounds directly imply results for the privacy notion studied by Kapralov and Talwar [KT13]. The notion is particularly relevant in a setting where we think of $A$ as a sum of rank 1 matrices each of bounded spectral norm.

**Definition E.4.** A randomized algorithm $M : \mathbb{R}^{d \times d'} \to R$ (where $R$ is some arbitrary abstract range) is $(\varepsilon, \delta)$-*differentially private under unit spectral norm changes* if for all pairs of matrices $A, A' \in \mathbb{R}^{d \times d'}$ satisfying $\|A - A'\|_2 \leqslant 1$, we have that for all subsets of the range $S \subseteq R$, the algorithm satisfies: $\Pr\{M(A) \in S\} \leqslant \exp(\varepsilon) \Pr\{M(A') \in S\} + \delta$.

**Lemma E.5.** *If* PPM *is executed with each* $G_\ell$ *sampled independently as* $G_\ell \sim N(0, \sigma^2)^{d \times p}$ *with* $\sigma = \varepsilon^{-1} \sqrt{4pL \log(1/\delta)}$, *then* PPM *satisfies* $(\varepsilon, \delta)$-*differential privacy under unit spectral norm changes.*

*If* $G_\ell$ *is sampled with i.i.d. Laplacian entries* $G_\ell \sim \text{Lap}(0, \lambda)^{n \times k}$ *where* $\lambda = 10 \varepsilon^{-1} pL \sqrt{d}$, *then* PPM *satisfies* $(\varepsilon, 0)$-*differential privacy under unit spectral norm changes.*

*Proof.* The first claim follows from the privacy proof in [HR12]. We sketch the argument here for completeness. Let $D$ be any matrix with $\|D\|_2 \leqslant 1$ (thought of as $A - A'$ in Definition E.4) and let $\|x\| = 1$ be any unit vector which we think of as one of the columns of $X = X_{\ell-1}$. Then, we have $\|Dx\| \leqslant \|D\| \cdot \|x\| \leqslant 1$, by definition of the spectral norm. This shows that the "$\ell_2$-sensitivity" of one matrix-vector multiplication in our algorithm is bounded by 1. It is well-known that it suffices to add Gaussian noise scaled to the $\ell_2$-sensitivity of the matrix-vector product in order to achieve differential privacy. Since there are $kL$ matrix-vector multiplications in total we need to scale the noise by a factor of $\sqrt{kL}$.

The second claim follows analogously. Here however we need to scale the noise magnitude to the "$\ell_1$-sensitivity" of the matrix-vector product which be bound by $\sqrt{n}$ using Cauchy-Schwarz. The claim then follows using standard properties of the Laplacian mechanism. ∎

Given the previous lemma it is straightforward to derive the following corollaries.

**Corollary E.6.** *Let* $A \in \mathbb{R}^{d \times d}$ *be a symmetric matrix with singular values* $\sigma_1 \geqslant \ldots \geqslant \sigma_d$ *and let* $\gamma = 1 - \sigma_{k+1}/\sigma_k$. *There is an algorithm that given a* $A$ *and parameter* $k$, *preserves* $(\varepsilon, \delta)$-*differentially privacy under unit spectral norm changes and outputs a rank* $2k$ *matrix* $B$ *such that with probability* $9/10$,

$$\|A - B\| \leqslant \sigma_{k+1} + \tilde{O}\left(\frac{\sigma_1 \sqrt{(k/\gamma) d \log d \log(1/\delta)}}{\varepsilon(\sigma_k - \sigma_{k+1})}\right).$$

*The* $\tilde{O}$-*notation hides the factor* $O\left(\sqrt{\log(\log(d)/\gamma)}\right)$.

*Proof.* The proof is analogous to the proof of Corollary E.3. ∎

A similar corollary applies to $(\varepsilon, 0)$-differential privacy.

**Corollary E.7.** *Let* $A \in \mathbb{R}^{d \times d}$ *be a symmetric matrix with singular values* $\sigma_1 \geqslant \ldots \geqslant \sigma_d$ *and let* $\gamma = 1 - \sigma_{k+1}/\sigma_k$. *There is an algorithm that given a* $A$ *and parameter* $k$, *preserves* $(\varepsilon, \delta)$-*differentially privacy under unit spectral norm changes and outputs a rank* $2k$ *matrix* $B$ *such that with probability* $9/10$,

$$\|A - B\| \leqslant \sigma_{k+1} + \tilde{O}\left(\frac{\sigma_1 k^{1.5} d \log(d) \log(d/\gamma)}{\varepsilon \gamma(\sigma_k - \sigma_{k+1})}\right).$$

*Proof.* We invoke PPM with $p = 2k$ and Laplacian noise with the scaling given by Lemma E.5 so that the algorithm satisfies $(\varepsilon, 0)$-differential privacy. Specifically, $G_\ell \sim \text{Lap}(0, \lambda)^{d \times p}$ where $\lambda = 10\varepsilon^{-1} pL\sqrt{d}$. Lemma A.3. Indeed, with probability $99/100$, we have

1. $\max_{\ell=1}^{L} \|G_\ell\| \leqslant O\left(\lambda \sqrt{kd} \log(kdL)\right) = O\left((1/\varepsilon\gamma) k^{1.5} d \log(d) \log(kdL)\right)$

2. $\max_{\ell=1}^{L} \|U^\top G_\ell\| \leqslant O\left(\lambda k \log(kL)\right) = O\left((1/\varepsilon\gamma) k^2 \sqrt{d} \log(d) \log(kL)\right)$

We can now plug these error bounds into Corollary 1.1 to obtain the bound

$$\|(I - X_L X_L^\top)U\| \leqslant O\left(\frac{k^{1.5} d \log(d) \log(d/\gamma)}{\varepsilon \gamma (\sigma_k - \sigma_{k+1})}\right)$$

Repeating the argument from the proof of Corollary E.3 gives the stated guarantee for low-rank approximation. ∎

The bound above matches a lower bound shown by Kapralov and Talwar [KT13] up to a factor of $\tilde{O}(\sqrt{k})$. We believe that this factor can be eliminated from our bounds by using a quantitatively stronger version of Lemma A.3. Compared to the upper bound of [KT13] our algorithm is faster by a more than a quadratic factor in $d$. Moreover, previously only bounds for $(\varepsilon, 0)$-differential privacy were known for the spectral norm privacy notion, whereas our bounds strongly improve when going to $(\varepsilon, \delta)$-differential privacy.

### E.3 Dimension-free bounds for incoherent matrices

The guarantee in Theorem 1.3 depends on the quantity $\|X_\ell\|_\infty$ which could in principle be as small as $\sqrt{1/d}$. Yet, in the above theorems, we use the trivial upper bound $1$. This in turn resulted in a dependence on the dimensions of $A$ in our theorems. Here, we show that the dependence on the dimension can be replaced by an essentially tight dependence on the *coherence* of the input matrix. In doing so, we resolve the main open problem left open by Hardt and Roth [HR13]. The definition of coherence that we will use is formally defined as follows.

**Definition E.8** (Matrix Coherence). We say that a matrix $A \in \mathbb{R}^{d \times d'}$ with singular value decomposition $A = U\Sigma V^\top$ has *coherence*
$$\mu(A) \overset{\text{def}}{=} \left\{d\|U\|_\infty^2, d'\|V\|_\infty^2\right\} .$$
Here $\|U\|_\infty = \max_{ij} |U_{ij}|$ denotes the largest entry of $U$ in absolute value.

Our goal is to show that the $\ell_\infty$-norm of the vectors arising in PPM is closely related to the coherence of the input matrix. We obtain a nearly tight connection between the coherence of the matrix and the $\ell_\infty$-norm of the vectors that PPM computes.

**Theorem E.9.** *Let $A \in \mathbb{R}^{d \times d}$ be symmetric. Suppose NPM is invoked on $A$, and $L \leqslant n$, with each $G_\ell$ sampled from $N(0, \sigma_\ell^2)^{d \times p}$ for some $\sigma_\ell > 0$. Then, with probability $1 - 1/n$,*

$$\max_{\ell=1}^{L} \|X_\ell\|_\infty^2 \leqslant O\left(\frac{\mu(A) \log(d)}{d}\right) .$$

*Proof.* Fix $\ell \in [L]$. Let $A = \sum_{i=1}^n \sigma_i u_i u_i^\top$ be given in its eigendecomposition. Note that

$$B = \max_{i=1}^{d} \|u_i\|_\infty \leqslant \sqrt{\frac{\mu(A)}{d}}.$$

We may write any column $x$ of $X_\ell$ as $x = \sum_{i=1}^{d} s_i \alpha_i u_i$ where $\alpha_i$ are non-negative scalars such that $\sum_{i=1}^{d} \alpha_i^2 = 1$, and $s_i \in \{-1, 1\}$ where $s_i = sign(\langle x, u_i \rangle)$. Hence, by Lemma E.13 (shown below), the signs $(s_1, \ldots, s_d)$ are distributed uniformly at random in $\{-1, 1\}^d$. Hence, by Lemma E.14 (shown below), it follows that $\Pr\left\{\|x\|_\infty > 4B\sqrt{\log d}\right\} \leqslant 1/n^3$. By a union bound over all $p \leqslant d$ columns it follows that $\Pr\left\{\|X_\ell\|_\infty > 4B\sqrt{\log d}\right\} \leqslant 1/d^2$. Another union bound over all $L \leqslant d$ steps completes the proof. ∎

The previous theorem states that no matter what the scaling of the Gaussian noise is in each step of the algorithm, so long as it is Gaussian the algorithm will maintain that $X_\ell$ has small coordinates. We cannot hope to have coordinates smaller than $\sqrt{\mu(A)/d}$, since eventually the algorithm will ideally converge to $U$. This result directly implies the theorem we stated in the introduction.

*Proof of Theorem 1.4.* The claim follows directly from Theorem 1.3 after applying Theorem E.9 which shows that with probability $1 - 1/n$,
$$\max_{\ell=1}^{L} \|X_\ell\|_\infty^2 \leqslant O\left(\frac{\mu(A) \log(d)}{d}\right) .$$
∎

### E.4 Proofs of supporting lemmas

We will now establish Lemma E.13 and Lemma E.14 that were needed in the proof of the previous theorem. For that purpose we need some basic symmetry properties of the QR-factorization. To establish these properties we recall the Gram-Schmidt algorithm for computing the QR-factorization.

**Definition E.10** (Gram-Schmidt). The *Gram-Schmidt orthonormalization* algorithm, denoted GS, is given an input matrix $V \in \mathbb{R}^{d \times p}$ with columns $v_1, \ldots, v_p$ and outputs an orthonormal matrix $Q \in \mathbb{R}^{d \times p}$ with the same range as $V$. The columns $q_1, \ldots, q_p$ of $Q$ are computed as follows:

For $i = 1$ to $p$ do:

  – $r_{ii} \leftarrow \|v_i\|$
  – $q_i \leftarrow v_i / r_{ii}$
  – For $j = i + 1$ to $p$ do:
    – $r_{ij} \leftarrow \langle q_i, v_j \rangle$
    – $v_j \leftarrow v_j - r_{ij} q_i$

The first states that the Gram-Schmidt operation commutes with an orthonormal transformation of the input.

**Lemma E.11.** *Let $V \in \mathbb{R}^{d \times p}$ and let $O \in \mathbb{R}^{d \times d}$ be an orthonormal matrix. Then,* $\mathrm{GS}(OV) = O \times \mathrm{GS}(V)$.

*Proof.* Let $\{r_{ij}\}_{ij \in [p]}$ denote the scalars computed by the Gram-Schmidt algorithm as specified in Definition E.10. Notice that each of the numbers $\{r_{ij}\}_{ij \in [p]}$ is invariant under an orthonormal transformation of the vectors $v_1, \ldots, v_p$. This is because $\|Ov_i\| = \|v_i\|$ and $\langle Ov_i, Ov_j \rangle = \langle v_i, v_j \rangle$. Moreover, The output $Q$ of Gram-Schmidt on input of $V$ satisfies $Q = VR$, where $R$ is an upper right triangular matrix which only depends on the numbers $\{r_{ij}\}_{i,j \in [p]}$. Hence, the matrix $R$ is identical when the input is $OV$. Thus, $\mathrm{GS}(OV) = OVR = O \times \mathrm{GS}(V)$. ∎

Given i.i.d. Gaussian matrices $G_0, G_1, \ldots, G_L \sim N(0,1)^{d \times p}$, we can describe the behavior of our algorithm by a deterministic function $f(G_0, G_1, \ldots, G_L)$ which executes subspace iteration starting with $G_0$ and then suitably scales $G_\ell$ in each step. The next lemma shows that this function is distributive with respect to orthonormal transformations.

**Lemma E.12.** *Let $f \colon (\mathbb{R}^{d \times p})^L \to \mathbb{R}^{n \times p}$ denote the output of PPM on input of a matrix $A \in \mathbb{R}^{n \times n}$ as a function of the noise matrices used by the algorithm as described above. Let $O$ be an orthonormal matrix with the same eigenbasis as $A$. Then,*

$$f(OG_0, OG_1, \ldots, OG_L) = O \times f(G_0, \ldots, G_L). \tag{4}$$

*Proof.* For ease of notation we will denote by $X_0, \ldots, X_L$ the iterates of the algorithm when the noise matrices are $G_0, \ldots, G_L$, and we denote by $Y_0, \ldots, Y_L$ the iterates of the algorithm when the noise matrices are $OG_0, \ldots, OG_L$. In this notation, our goal is to show that $Y_L = OX_L$.

We will prove the claim by induction on $L$. For $L = 0$, the base case follows from Lemma E.11. Indeed,

$$Y_0 = \mathrm{GS}(OG_0) = O \times \mathrm{GS}(G_0) = OX_0.$$

Let $\ell \geqslant 1$. We assume the claim holds for $\ell - 1$ and show that it holds for $\ell$. We have,

$$
\begin{aligned}
Y_\ell &= \mathrm{GS}(AY_{\ell-1} + OG_\ell) \\
&= \mathrm{GS}(AOX_{\ell-1} + OG_\ell) && \text{(by induction hypothesis)} \\
&= \mathrm{GS}(O(AX_{\ell-1} + G_\ell)) && \text{($A$ and $O$ commute)} \\
&= O \times \mathrm{GS}(AX_{\ell-1} + G_\ell) && \text{(Lemma E.11)} \\
&= OX_\ell.
\end{aligned}
$$

Note that $A$ and $O$ commute, since they share the same eigenbasis by the assumption of the lemma. This is what we needed to prove. ∎

The previous lemmas lead to the following result characterizing the distribution of signs of inner products between the columns of $X_\ell$ and the eigenvectors of $A$.

**Lemma E.13** (Sign Symmetry). *Let $A$ be a symmetric matrix given in its eigendecomposition as $A = \sum_{i=1}^{d} \lambda_i u_i u_i^\top$. Let $\ell \geqslant 0$ and let $x$ be any column of $X_\ell$, where $X_\ell$ is the iterate of PPM on input of $A$. Put $S_i = sign(\langle u_i, x \rangle)$ for $i \in [d]$. Then $(S_1, \ldots, S_d)$ is uniformly distributed in $\{-1, 1\}^d$.*

*Proof.* Let $(z_1, \ldots, z_d) \in \{-1, 1\}^d$ be a uniformly random sign vector. Let $O = \sum_{i=1}^{d} z_i u_i u_i^\top$. Note that $O$ is an orthonormal transformation. Clearly, any column $Ox$ of $OX_\ell$ satisfies the conclusion of the lemma, since $\langle u_i, Ox \rangle = z_i \langle u_i, x \rangle$. Since the Gaussian distribution is rotationally invariant, we have that $OG_\ell$ and $G_\ell$ follow the same distribution. In particular, denoting by $Y_\ell$ the matrix computed by the algorithm if $OG_0, \ldots, OG_\ell$ were chosen, we have that $Y_\ell$ and $X_\ell$ are identically distributed. Finally, by Lemma E.12, we have that $Y_\ell = OX_\ell$. By our previous observation this means that $Y_\ell$ satisfies the conclusion of the lemma. As $Y_\ell$ and $X_\ell$ are identically distributed, the claim also holds for $X_\ell$. ∎

We will use the previous lemma to bound the $\ell_\infty$-norm of the intermediate matrices $X_\ell$ arising in power iteration in terms of the coherence of the input matrix. We need the following large deviation bound.

**Lemma E.14.** *Let $\alpha_1, \ldots, \alpha_d$ be scalars such that $\sum_{i=1}^{d} \alpha_i^2 = 1$ and $u_1, \ldots, u_d$ are unit vectors in $\mathbb{R}^n$. Put $B = \max_{i=1}^{d} \|u_i\|_\infty$. Further let $(s_1, \ldots, s_d)$ be chosen uniformly at random in $\{-1, 1\}^d$. Then,*

$$\Pr\left\{ \left\| \sum_{i=1}^{d} s_i \alpha_i u_i \right\|_\infty > 4B\sqrt{\log d} \right\} \leqslant 1/d^3 .$$

*Proof.* Let $X = \sum_{i=1}^{d} X_i$ where $X_i = s_i \alpha_i u_i$. We will bound the deviation of $X$ in each entry and then take a union bound over all entries. Consider $Z = \sum_{i=1}^{d} Z_i$ where $Z_i$ is the first entry of $X_i$. The argument is identical for all other entries of $X$. We have $\mathbb{E}\, Z = 0$ and $\mathbb{E}\, Z^2 = \sum_{i=1}^{d} \mathbb{E}\, Z_i^2 \leqslant B^2 \sum_{i=1}^{d} \alpha_i^2 = B^2$. Hence, by Theorem A.1 (Chernoff bound),

$$\Pr\left\{ |Z| > 4B\sqrt{\log(d)} \right\} \leqslant \exp\left( -\frac{16B^2 \log(d)}{4B^2} \right) \leqslant \exp(-4\log(d)) = \frac{1}{d^4} .$$

The claim follows by taking a union bound over all $d$ entries of $X$. ∎