[Reviews · NeurIPS 2014]

Submitted by Assigned_Reviewer_31

The paper conveys a broad analysis of the convergence of the pertubed power method, called noisy power method (NPM), recently applied in a number of applications, which subsume computation of singular vectors of a matrix in a presence of noise. The authors infer and state an explicit general convergence rate of NPM subject to the noise of a bounded magnitude. Thereby, they ensure a global convergence of NPM, only local convergence of NPM being previously claimed for the matrix completion application. In the case of the streaming PCA, the generality of the estimates allows to extend the model to arbitrary distributions. Regarding the privacy-preserving spectral analysis, a fast NPM based algorithm is elaborated, which keeps a nearly optimal accuracy.
Quality and Clarity: The paper is technically sound. The presentation of the developed theory is clear and cohesive.
Originality: This work extends and extensively generalizes earlier applications of NPM imposing new theoretical findings regarding the convergence of the method.
Significance: The work develops an important and valuable theory that has already shown to advance the applications of NPM.
Summary: It is a solid work that establish theoretical grounds for the noisy power method.

Submitted by Assigned_Reviewer_35

This a a good paper about a generalized formulation and analysis of several randomized linear algebra approximations, such as streaming PCA and privacy aware spectral analysis. The paper is very clean and nicely written, it does not contain empirical experiments, which is not a problem since it is mainly a theoretical contribution. The paper is clean in its present form, and don't have detailed comments for it.
Summary: A good paper that gives a general formulation for randomized linear algebra problems, with interesting theoretical results.

Submitted by Assigned_Reviewer_39

The paper presents a general form for the noisy power method (NPM) and uses this as a framework to study some interesting special cases including error bounds both in general and for these cases. The NPM is the extension of the well known power method for computing the dominant singular vector.

This work provides a general form that encompasses a number of interesting and useful sub-problems. Among these the streaming PCA and privacy-preserving spectral analysis are discussed at length.

By unifying these methods under a single general form and providing error bounds that can be used in both the special and general cases this work provides a strong contribution.
Summary: An interesting and welcome unification of different forms and bounds on the noisy power method. This provides good insight and useful bounds that can be applied more widely.
Author Feedback
Author feedback is not available.